# Exploiting natural chemical photosensitivity of anhydrotetracycline and tetracycline for dynamic and setpoint chemo-optogenetic control

Armin Baumschlager [1], Marc Rullan[1] & Mustafa Khammash[1] ✉

The transcriptional inducer anhydrotetracycline (aTc) and the bacteriostatic antibiotic tetracycline (Tc) are commonly used in all fields of biology for control of transcription or translation. A drawback of these and other small molecule inducers is the difficulty of their removal from cell cultures, limiting their application for dynamic control. Here, we describe a simple method to overcome this limitation, and show that the natural photosensitivity of aTc/Tc can be exploited to turn them into highly predictable optogenetic transcriptional- and growth-regulators. This new optogenetic class uniquely features both dynamic and setpoint control which act via population-memory adjustable through opto-chemical modulation. We demonstrate this method by applying it for dynamic gene expression control and for enhancing the performance of an existing optogenetic system. We then expand the utility of the aTc system by constructing a new chemical bandpass filter that increases its aTc response range. The simplicity of our method enables scientists and biotechnologists to use their existing systems employing aTc/Tc for dynamic optogenetic experiments without genetic modification.

[1] Department of Biosystems Science and Engineering (D-BSSE), ETH—Zürich, Mattenstrasse 26, 4058 Basel, Switzerland.
✉email: mustafa.khammash@bsse.ethz.ch

L ight regulation of cellular functions (optogenetics) has become an intensively studied field in recent years[1–5] due to its immense potential for precise spatial and temporal regulation. Light has been used to regulate protein function, membrane recruitment, nuclear localization, and gene expression among other applications[1]. Optogenetics allows one to override the endogenous control of cellular processes using fast changeable light inputs. These fast-changing inputs enable dynamic perturbations that can provide new insight into the organizing principles of biology and the study of gene networks[6], allowing for new approaches for metabolic engineering[7–11]. In synthetic biology, the use of light as a computer–cell interface for both input and output has enabled novel cybergenetic regulation schemes, whereby desired regulatory functions can be implemented in silico inside a computer, obviating the need to implement complex genetic regulatory circuits[12,13]. In these applications, protein-based optogenetic regulators, such as light-inducible polymerases[14–16], transcription factors[17], and two-component systems[18–20] have been developed to fulfill such dynamic gene

expression requirements. However, these regulators require continuous or reoccurring light inputs (pulses) because activated protein regulators are diluted out in growing cells or inactivated through their inherent dark-state reversion that has to be fast to allow for rapid adjustments. This also implies that the light input has to be applied at an appropriate frequency to maintain expression at desired levels (Fig. 1a left).

Chemical approaches to light control of transcription rely primarily upon the incorporation of light-sensitive moieties such as UV-responsive nitrobenzyl groups into the scaffolds of small molecules, such as arabinose[21], abscisic acid[22], and numerous other bioactive molecules, including siRNAs[1]. Such approaches make small-molecule-induced gene expression systems, which are a key component in synthetic biology[23] and biotechnological applications[24], light-inducible and open them up to novel applications, as classical chemical inducers are limited in their application in space and time. However, one major drawback of these light-activatable small molecules is their irreversibility, which means that these molecules can only be activated once, and it is

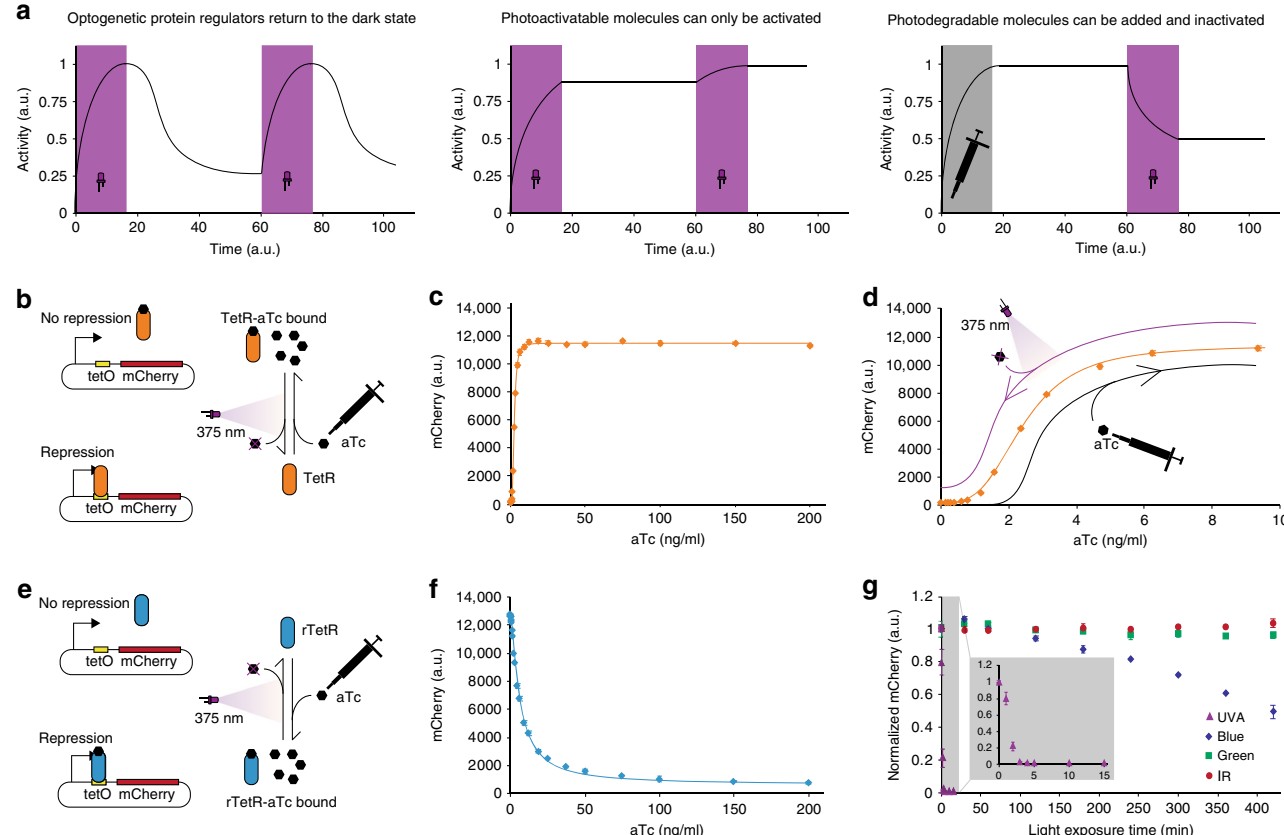

**Fig. 1 Photosensitivity of anhydrotetracycline. a** Photodegradation of molecules provides the unique feature of single-input setpoint control to increase or decrease the activity of the controlled system in comparison to optogenetic proteins or photoactivatable molecules. TetR (orange diamonds and lines, **b–d**) and rTetR (blue diamonds and line, **e**, **f**)-controlled expression of mCherry is titratable with aTc (**c**, **d**, **f**). **b** Addition of aTc to cells containing the TetR repressor leads to unbinding from *tetO* operator sequences allowing transcription from the promoter. No aTc or aTc inactivation allows for TetR binding to *tetO* and repression of transcription. mCherry was used as a fluorescent reporter to visualize expression from a TetR-controlled promoter. **c** Dose–response curve of aTc concentration and mCherry expression for TetR-controlled gene expression and model fit using Eq. (1). **d** Dose–response curve as shown in **c**, zoomed in on the aTc concentration range that is sufficient for adjusting the expression level of TetR-controlled gene expression from its minimal to the maximal value. UVA illumination and addition of aTc can be used to change the concentration of aTc in medium containing cells, effectively allowing one to adjust the expression level along the dose–response curve of aTc. **e** Addition of aTc to cells containing the rTetR repressor, which works opposite to TetR, leads to binding to *tetO* operator sequences. Accordingly, no aTc or aTc inactivation leaves rTetR unbound from *tetO* that results in expression of mCherry. **f** Dose–response curve of aTc concentration and mCherry expression for rTetR-controlled gene expression and model fit using Eq. (3). **g** Incubation of medium containing aTc with different wavelengths of light shows fast degradation dynamics with UVA light (**g**, gray inset) compared to slow degradation with blue, and no observable degradation with green and infrared light measured through TetR-controlled mCherry expression (**g**). Diagrams show mean mCherry expression values and standard deviation of three biological replicates (n = 3) measured after 5 h incubation time in all cases. Source data are provided as a Source data file.

not possible to return them to an inactive state (Fig. 1a middle). This makes them unsuitable for dynamic regulation in scenarios, such as batch or fed-batch fermentation processes, where the removal of inducer molecules from the medium is not possible. In addition, photocaged molecules are not commercially available, so their chemical synthesis is limited to a small number of specialized labs, which increases their cost and sets a significant hurdle to their use in optogenetic experiments.

Although light, in contrast to chemical induction methods, has the advantage of imposing negligible effects on the experimental conditions, phototoxicity of cells has been reported for high intensity light exposure of different organisms[25]. This can be attributed to absorption of chemical moieties, such as conjugated pi systems that transform a molecule into a toxic product. However, the reverse (e.g., the phototransformation of a toxic substrate into a nontoxic product) is also possible and was described for detoxification of tetracycline (Tc) in the environment[26]. Tcs are bacteriostatic polyketide antibiotics that bind to the 30 S subunit of the bacterial ribosome and contain such light-sensitive chemical moieties. Binding of ribosomes inhibits cellular translation resulting in growth inhibition[27]. As one mechanism of resistance against this antibiotic, bacteria have evolved the protein-sensor TetR, which functions as a transcriptional repressor. TetR binds specific operator sequences (tetO) of the tet-promoter in the absence of Tc and represses transcription of the antibiotic resistance. Tc is bound by TetR, causing a conformational change of the repressor molecule, which leads to its release from the operator sequence and subsequent expression of the antibiotic resistance[27]. Both the antibiotic resistance gene for plasmid maintenance and the Tc-inducible expression system are commonly used in all fields of biology, and are inexpensive commercially available chemicals.

Tc derivatives such as anhydrotetracycline (aTc) or doxycycline (DOX) are used in gene expression systems because of their improved induction profiles, reduced toxicity, and improved chemical and biological half-life[28,29]. aTc binds TetR more efficiently than Tc and has lower antibiotic activity against *Escherichia coli*[28]. Moreover, it was suggested that aTc acts at a different level from Tc and does not interfere with the translational process[28]. Together with LacI-, AraC-, and LuxR-regulated promoters, and their respective inducer molecules isopropyl β-D-1-thiogalactopyranoside, L-arabinose, and N-(3-oxohexanoyl)-L-homoserine lactone, these small-molecule-induced regulation systems are the dominant components for transcriptional control in synthetic biology and biotechnological applications[24,30,31]. They provide advantageous features, such as a high dynamic range, low leakiness, and a robust stable output.

As the persistent presence of Tc promotes the development of microbial antibiotic resistance, and can lead to acute and chronic toxicity to organisms[26], its unmodified release into the environment is highly problematic. Therefore, extensive research has focused on studying and enhancing the degradation of tetracyclines, because of their wide use as antibiotics in veterinary medicine and farming. To detoxify tetracyclines in contaminated soil, phototransformation was identified as a promising method that is caused by the ability of conjugated benzene-enone structures of tetracyclines to absorb light in the ultraviolet–visible wavelength spectrum[26]. In aqueous solution, the absorbance peak of Tc lies between 357–372 nm, depending on the pH[26]. In this work, we apply these findings to biological research and challenge the perception that instability of biofunctional chemicals is an unwanted effect for robust predictable control[32]. Moreover, we exploit such instability, namely photosensitivity of the translation inhibitor Tc and the transcriptional inducer aTc, and use it to transform these classical static systems into dynamic controllers of their respective biological functions (Fig. 1a right).

## Results

**Activation and repression of transcription with aTc.** To characterize the light-dependent degradation of aTc, we used a bacterial strain that serves as a biosensor/reporter for aTc concentration. This strain constitutively expresses TetR from the chromosome, and contains a plasmid that encodes the red fluorescent protein mCherry under control of a promoter that is regulated via the TetR-binding site tetO (Fig. 1b). TetR binds tetO in the absence of aTc and represses the expression of mCherry. This repression is weakened with increasing concentrations of aTc, allowing for titratable expression of a gene of interest (Fig. 1c, d). The response was measured at the single-cell level through flow cytometry. All mCherry fluorescence values in this work represent steady-state expression.

We also used the TetR moiety of the transactivator rtTA3 (amino acids 1–207), including the mutations that reverse TetR function[33]. This reversed Tet repressor (rTetR) binds tetO not in the absence, but in the presence of Tc, allowing for small-molecule-induced repression of transcription (Fig. 1e). We expressed this regulator from a synthetic araB-promoter that can be induced with arabinose, but is not repressed with glucose[14]. Basal expression of rTetR from this promoter is sufficient for aTc-inducible expression. rTetR-binding is titratable with increasing concentration of aTc (Fig. 1f).

**Wavelength-dependent light sensitivity of aTc.** We first experimentally characterized aTc photosensitivity using four different wavelength LEDs (UVA: 375 nm, blue: 472 nm, green: 525 nm, and infrared: 740 nm) with their respective maximal intensity in a 24-well light plate[34]. A concentration of 10 ng/ml aTc, which is not saturating for our TetR strain, was used in order to detect any decrease in the aTc concentration. Addition of an equal volume of media containing cells after light-induced aTc inactivation, resulted in a concentration of 5 ng/ml aTc. We observed that aTc is readily inactivated with UVA light (complete inactivation after 3 min), and slowly inactivated with blue light (50% inactivation after 7 h). No measurable inactivation was observed with green or infrared light in our experimental time frame of 7 h (Fig. 1g). This also resembles recently reported UV–vis spectra of Tc, which show absorption peaks in aqueous solution of 360–375 nm and low absorption for wavelengths > 500 nm (ref. [26]). We therefore concluded that UVA light effectively inactivates aTc in timescales (seconds to minutes) that are on par with other fast optogenetic systems[35].

**Controlled aTc concentration change with UVA illumination.** We further characterized the degradation dynamics of aTc in response to illumination duration with fixed light intensities ($4.1 \pm 0.5$ mW/cm$^2$), as well as with varying light intensities, given a fixed duration (4 min). We again used nonsaturating aTc concentrations (10 ng/ml for our TetR strain and 25 ng/ml for our rTetR strain) to detect any aTc concentration changes. We found that with full intensity of conventional UVA LEDs, a TetR system can be inactivated within 3 min from nearly full induction to full repression (Fig. 2a left). Although aTc degradation is fast, intermediate expression levels can be achieved through short illumination durations at full intensity (Fig. 2a left), or reduced light intensities (Fig. 2a right). Similarly, aTc inactivation led to UVA-induced activation of transcription using our rTetR system (Fig. 2b). Single-cell analysis revealed that all measured expression levels show a very narrow fluorescence distribution, as all cells are exposed to the same concentration of aTc in the medium (Supplementary Figs. 1 and 2). This characterization also shows that the degradation mechanism can be used to reduce once-added aTc concentrations to required levels, allowing for dynamic

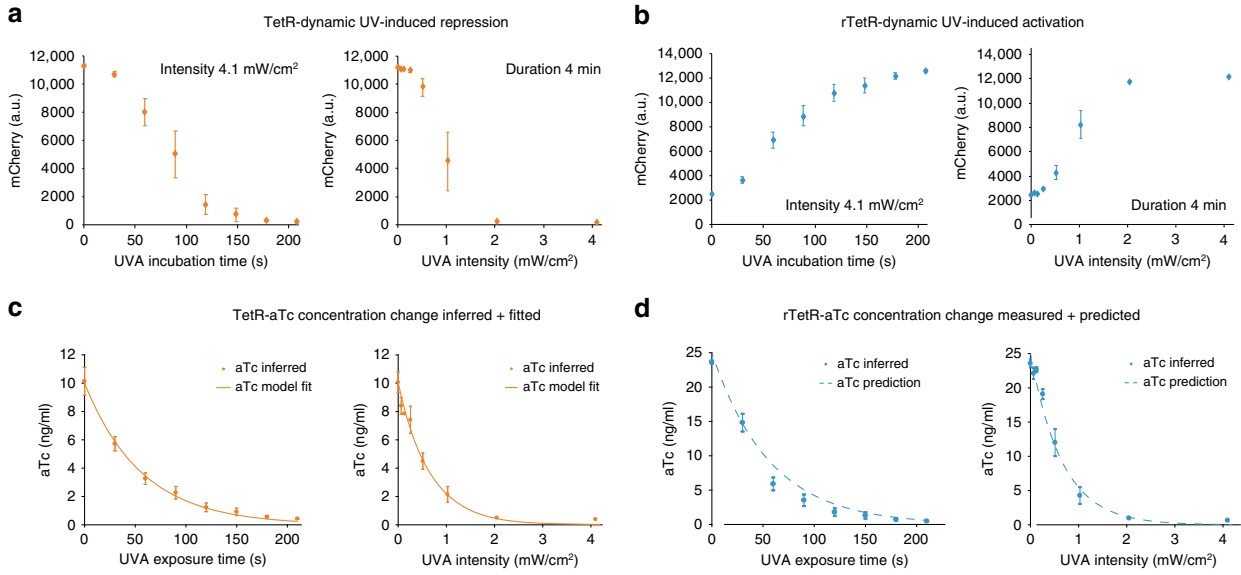

**Fig. 2 UVA mediated aTc inactivation.** aTc inactivation was studied using TetR (orange diamonds and lines, **a**, **c**) and rTetR (blue diamonds and dashed lines, **b**, **d**)-controlled promoters as biosensors. aTc inactivation is controllable with duration (**a**, **b** left, intensity 4.1 mW/cm²) and intensity (**a**, **b** right, duration 4 min) of the UVA light input. **a** Left, using 10 ng/ml aTc and a TetR containing strain, we applied UVA light with an intensity of 4.1 mW/cm² with increasing duration (0, 30, 60, 90, 120, 150, 180, and 210 s) to different samples to follow aTc degradation over time. **a** Right, using again a TetR containing strain and 10 ng/ml aTc, we applied increasing UVA light intensities (0.064, 0.13, 0.26, 0.51, 1.0, 2.1, and 4.1 mW/cm²) at a fixed duration of 4 min to follow light-intensity-dependent aTc degradation. **b** Left, analogous to the experiment shown in **a**, left, we also performed duration-dependent aTc degradation with our rTetR containing strain and 25 ng/ml aTc. **b** Right, analogous to **a**, right, we incubated our rTetR containing strain and 25 ng/ml aTc with increasing UVA light intensities. **c** The inferred aTc concentrations of the duration and intensity degradation experiments shown in **a** were fit to an exponential decay model Eq. (2) using the TetR-controlled mCherry expression system as a sensory module. The exponential decay equations are of the form $f(u) = e^{-cu}$, where $u$ represents the input (light intensity or light duration), and $c$ is the decay factor. **d** Using the same degradation model and parameters obtained from the model fitting shown in **c**, aTc degradation is correctly predicted in the rTetR-controlled system. Experiments were performed in nonsaturating aTc concentrations with 10 ng/ml for the TetR and 25 ng/ml for the rTetR containing strain. Diagrams show mean mCherry expression values and standard deviation of three biological replicates ($n = 3$) measured after 5 h incubation time in all cases. Source data are provided as a Source data file.

control by adding aTc and removing it through UVA illumination (Fig. 1a right, Fig. 1d).

We further investigated if UVA or the inactivation of aTc under UVA illumination induces cell damage. For this, we performed a set of experiments with long UVA illumination durations, and high aTc concentrations. Both the illumination time as well as the aTc concentrations dramatically exceed the time needed for inactivation of conventionally used aTc concentrations as previously discussed (Fig. 2a, b), and show that UVA exposure of up to 30 min with full intensity in our setup shows negligible effects on cell growth (Supplementary Fig. 3a–c). Further, we tested if also high concentrations of aTc can be inactivated. We found that our method allows for inactivation of up to 500 ng/ml aTc (Supplementary Figs. 3d, e and 4), which enables ~100 cycles of practically full induction of TetR-based systems (5 ng/ml shows close to full induction for TetR, Fig. 1d) to uninduced cells.

We then investigated if aTc inactivation can also be achieved in denser cell cultures than the logarithmic growth cultures we used for all characterization experiments, as bacterial cultures absorb visible light. Our results in Supplementary Fig. 5 indeed show that aTc inactivation is possible with highly turbid cultures. As in all other optogenetic methods, the method needs to be calibrated on the specific experimental setup used (cell culture density, media composition, light application path, …) if degradation to precise intermediate aTc levels is required. If, however, on-off control is required, UVA light can be applied in excess to inactivate all inducer molecules. As an example for simple on-off control, we

used our method as a single-inducer toggle switch by adding aTc to a concentration of 7.5 ng/ml, and removed it through a 3 min UVA light pulse at maximal intensity repeatedly for 12 consecutive setpoints (Supplementary Fig. 6).

The controllability of aTc inactivation in our setup further motivated us to mathematically model the light-dependent aTc degradation, so we can determine beforehand how much aTc must be added or which light input must be applied to achieve a predefined expression-level. For this, we fitted the TetR aTc dose–response to a Hill equation (Eq. (1), Fig. 1c), and then used the measured aTc degradation in response to light intensity and duration to fit an equation for exponential decay (Eq. (2), Fig. 2c). Therefore, the aTc inactivation is modeled independently from the sensory output module, which in this case is the TetR-controlled mCherry expression. We then also fitted the aTc dose–response curve of rTetR to a Hill equation for repression (Eq. (3), Fig. 1f). Using the fitted mathematical model for light duration- and intensity-dependent aTc degradation with TetR as a sensor, we predicted the response of the rTetR system to a given light input. The matching predictions of the aTc inactivation (Fig. 2d) using the rTetR system from model fits of aTc degradation using the TetR system, which has opposite functionalities and has different aTc sensitivities, shows that aTc degradation is highly predictable. If, however, a stable expression level of a TetR-based system is required, the inducer DOX can be used in settings in which the culture is exposed to light. DOX shows dramatically improved UVA stability when incubated with full intensity light (4.1 mW/cm²) for 15 min (Supplementary Fig. 7).

**Dynamic gene expression via aTc addition and inactivation**. To demonstrate how dynamic expression control in batch cultures with aTc as inducer could be applied in practice, we performed a time-course experiment, in which the aTc concentration was changed multiple times during the course of the experiment by alternating addition and UVA inactivation of the chemical in a batch culture (Fig. 3a). The changes in the aTc concentration are schematically illustrated in Fig. 3a left, and the corresponding measured experimental data is shown in Fig. 3a right. aTc-free medium was inoculated with cells expressing TetR, which again controls expression of mCherry (Fig. 3a; condition 1). We then added aTc to a concentration of 5 ng/ml (condition 2), before partial inactivation through 60 s UVA illumination (condition 3) and then further inactivation with a second UVA pulse of 30 s (condition 4). We then added aTc to 10 ng/ml to reach close to full induction (condition 5), and lastly completely inactivated aTc in the medium with a UVA pulse of 5 min (condition 6) to reach the basal expression level seen before addition of aTc (condition 1). The setpoints shown are steady-state expression levels that result from the consecutively applied inputs. This illustrates the unique property of this optogenetic control that allows for the application of complex inputs that are maintained until a new input is applied.

**Dual optogenetic control to reduce dark state expression**. As another practical example, we show that dynamic aTc regulation can be used to reduce the dark-state activity of protein-based optogenetic systems by implementing two-stage optogenetic control. For this, we used a recently developed blue light-inducible T7 DNA-dependent RNA polymerase, Opto-T7RNAP* (563)[14]. It consists of two components that we placed under the control of aTc-inducible promoters. We further added a *tetO* site downstream of its corresponding T7 promoter (Fig. 3b left) and exchanged *lacI* to *rtetR* under control of the wild-type lac promoter. Due to the stronger expression of the *tet-*, compared to the *araB*-promoter used in the original study[14], this system shows a significantly increased dark state expression and a light-induced expression that causes growth defects in our testing cells. This leads to a reduced fold induction of ~65-fold comparing dark and light-induced expression, which is significantly lower than the ~300-fold of the original Opto-T7RNAP system. However, aTc can be used to lower the dark state expression by reducing the expression of Opto-T7RNAP components, as well as inhibiting the expression from the T7 promoter, which leads to an increase of the dark and light-induced expression to ~700-fold (Fig. 3b right). This shows that this system can also be used as an additional knob to combine with, and improve, existing optogenetic systems.

**Transient translation inhibition with Tc**. Given that Tc acts as a bacteriostatic antibiotic, we hypothesized that growth of bacterial cells can be steered through addition and light-induced inactivation of the antibiotic. This should allow for dynamic control of the transition of cells from a translationally inactive "stationary phase" to an active growing phase (Fig. 3c left). We first experimentally determined the minimum inhibitory concentration for our experimental setup to be 400 ng/ml. We then used this concentration of Tc in medium illuminated with increasing UVA duration at $4.1 \pm 0.5$ mW/cm$^2$ and 37 °C. Adding the same amount of cells to these samples, as well as to controls without Tc and with 10 µg/ml Tc (a commonly used concentration in selection media), we show that: (1) Tc can be predictably degraded with UVA light; and (2) that decreasing Tc concentrations caused through UVA inactivation result in increasing growth rates, which makes this system applicable for growth

control (Fig. 3c right). We also performed the same experiment in the presence of cells and confirmed the functionality (Supplementary Fig. 8). While we intended Tc as an example that light-induced inactivation of biofunctional chemicals is not limited to aTc and gene expression, future characterization of Tc inactivation might include findings from environmental and industrial Tc detoxification processes, which is a well-studied subject[26,36–38].

**TetR–rTetR chemical bandpass filter for aTc induction**. Desired features of an expression system are that on the one hand it is only active upon inducer application, and on the other hand it is easily controllable to intermediate expression levels. The first property allows construction (e.g., cloning of constructs, transformation of strains) and growth of cells without expression of a gene of interest, as the expression system is not active without inducer presence. The second feature enables one to precisely optimize expression levels for a given purpose. While the first requirement is met with TetR-regulated promoters that are only activated once aTc is added, the second property is not fulfilled due to the steep dose–response curve where small changes in aTc concentration change the output dramatically (Fig. 1c). While this does not pose a problem for UVA light-induced deactivation, which we show can be modeled and applied precisely, it might become significant for longer-term cultivation of cells when the half-life of aTc (15 h for 37 °C and 30 h for 30 °C (ref. [32])) becomes a factor that needs to be considered. rTetR-regulated promoters fulfill this latter requirement, as the inversed dose–response curve is significantly less steep (Fig. 1f), therefore small changes in aTc concentration do not have such a high impact on the expression level. However, this system shows maximum expression without inducer presence causing the aforementioned limitations. Therefore, a combination of both systems, which does not express in the absence of inducer, but is not sensitive to small changes in the inducer concentration is desirable. We achieve this by combining TetR and rTetR in the same cell, and practically constructing a chemical bandpass filter (Fig. 4a left). An empirical mathematical model for the bandpass dose–response was constructed in Eqs. (4) and (5) through combining the dose–response models of TetR and rTetR described in Eqs. (1) and (3), respectively (Supplementary Fig. 9). Without aTc, TetR binds the promoter and shuts down transcription. This binding is completely released from the promoter with aTc concentrations > 10 ng/ml (Fig. 1d). However, the effective range of aTc concentrations for rTetR spans until 200 ng/ml (Fig. 1f), allowing for adjustment of the expression level through aTc addition or its light inactivation (Fig. 4a right). In addition, this circuit only requires one promoter and two repressor proteins with opposing function, making it a much simpler implementation of a chemical bandpass filter compared to previous work[39–44].

As a second example, we use the same setup, but apply this bandpass filter to a T7RNAP expression system (Fig. 4b left). Combining this strong T7RNAP expression system with a plasmid that contains *mCherry* under control of the T7 promoter, and transforming it into cells can lead to mutations that inactivate the burdensome constitutive expression of the fluorescent protein. The bandpass filter allows for the construction and maintenance of the strain without this burden in the absence of aTc, but still enables nicely tunable expression through aTc concentration change (Fig. 4b right).

**Discussion**

In summary, we describe a new approach for optogenetic inactivation of natural molecules that enables dynamic single-input setpoint control, and then we show that photosensitivity of

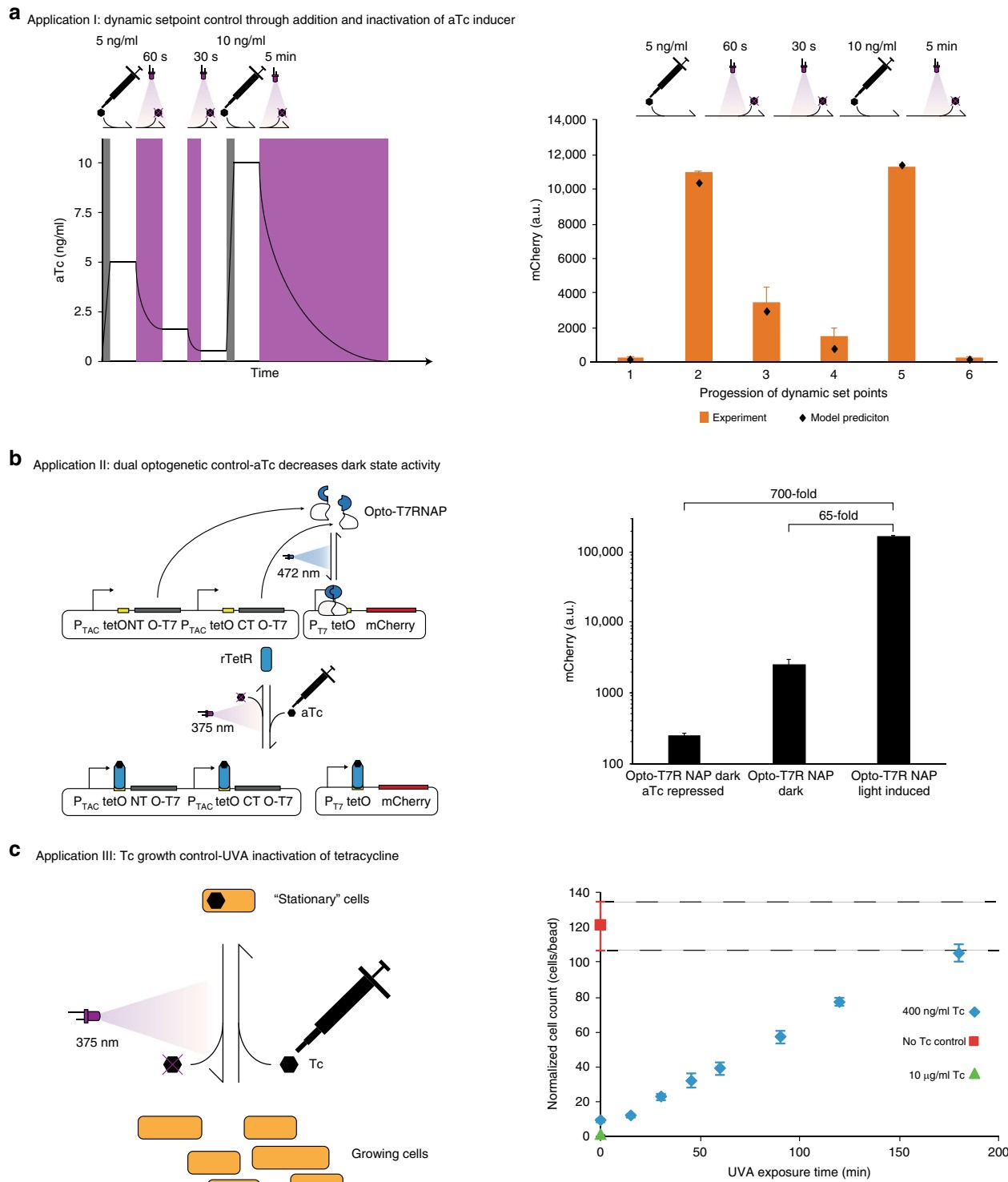

**Fig. 3 Applications of aTc and Tc photodegradation. a** Dynamic setpoint control through mCherry expression via addition of aTc and its removal through UVA. Schematic illustration (**a** left) of aTc concentration change in the dynamic experiment (**a** right) through the aTc addition and UVA degradation inputs. Cells expressing TetR which again controls expression of mCherry without aTc (condition 1), after addition of aTc to 5 ng/ml (condition 2), partial aTc removal through 60 s UVA illumination (condition 3), further removal through a second UVA pulse of 30 s (condition 4), then addition of aTc to 10 ng/ml (condition 5), and finally complete aTc removal with a UVA pulse of 5 min (condition 6). Diagrams show mean mCherry expression values and standard deviation of three biological replicates in orange bars and black diamonds for mathematical predictions, using our parameterized model. **b** Dual optogenetic control of *mCherry* expression. rTetR is used to reduce dark state expression of Opto-T7RNAP* (563) in the presence of aTc (**b** left), effectively increasing the dark to light fold change of the optogenetic system from ~65- to ~700-fold (**b** right). **c** Light-inducible control of growth rate with Tc. Tc can be added directly and inactivated with UVA to shift cells between "stationary" and growing cells (**c** left). The antibiotic is inactivated through UVA illumination (4.1 mW/cm²) over time, which leads to intermediate growth rates (**c** right) or complete removal of the antibiotic with growth rates comparable to the control without Tc. Diagrams show mean values and standard deviation of three biological replicates ($n = 3$) of mCherry values for gene expression experiments, and cell counts normalized to counting beads for growth experiments measured after 5 h incubation time in all cases. Source data are provided as a Source data file.

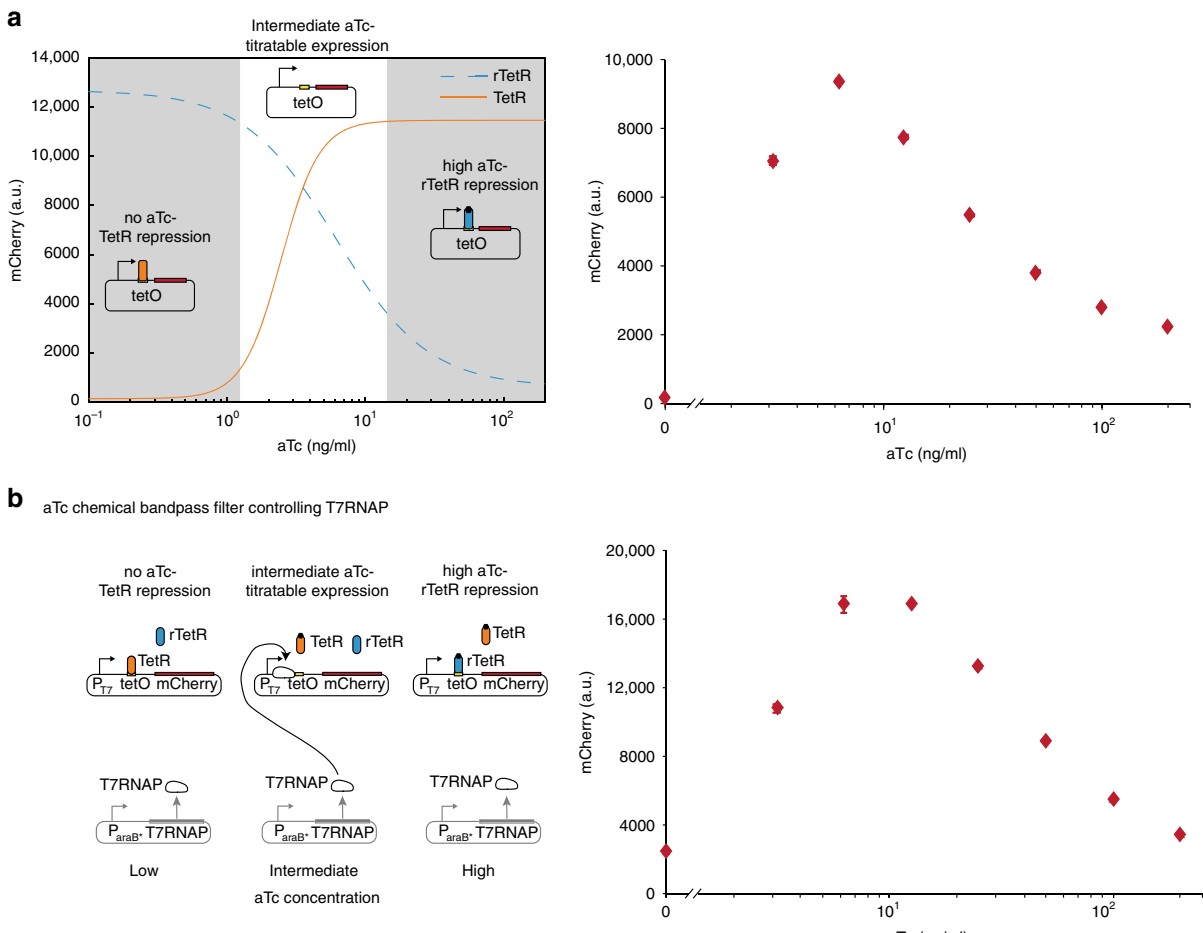

**Fig. 4 aTc concentration bandpass filter. a** Left, combining TetR (orange line) and rTetR (blue dashed line) in the same cell enables activation of aTc-induced expression due to the difference in effective inducer concentration. No aTc causes TetR repression and intermediate aTc concentrations (~10–200 ng/ml) enable titratable mCherry expression (**a** right). **b** aTc bandpass filter controlling the T7RNAP expression system. Expanding the topology of **a**, *mCherry* expression through T7RNAP R632S W727C is controlled by the bandpass filter (**b** left). Again, no aTc causes TetR repression and intermediate aTc concentrations (~10–200 ng/ml) enable titratable mCherry expression via the T7RNAP (**b** right). Diagrams show mean mCherry expression values and standard deviation of three biological replicates (*n* = 3) measured after 5 h incubation time in all cases. Source data are provided as a Source data file.

molecules can be effectively exploited for dynamic control of cellular functions. We demonstrate this with the small molecule inducer aTc for transcriptional, and the bacteriostatic antibiotic Tc for translational control. We show that this novel optogenetic regulation principle allows for dynamic setpoint control in batch cultures through a single input via the addition or optogenetic removal of molecules, which yields a novel modality of control that is not possible with current protein-based or photocaged optogenetic regulators.

Although our experiments were performed in the bacterium *E. coli*, this mechanism could also be applied to other organisms as Tc-controlled gene expression has been used in a variety of eukaryotic systems, such as plants, yeast, and mammalian cells as repressors, as well as activators[33,45]. Further, since green and infrared light has no effect and blue light inactivates aTc slowly, these wavelengths could also potentially be multiplexed with other standard protein-based optogenetic systems that activate/inactivate with these wavelengths. We further show how aTc photosensitivity can be used to reduce dark-state activity of protein-based optogenetic regulators and create an aTc chemical bandpass filter that improves application of optogenetic, as well as classical aTc-induced expression systems. In addition, we control growth through UVA-induced degradation of another chemical,

the bacteriostatic antibiotic Tc, to transition inhibited "stationary cells" into growing cells. This shows the feasibility of controlling different cellular functions (transcription and translation) using our proposed approach. Finally, this work may well lead to dynamic applications for other light-sensitive molecules, which are numerous in biology, ranging from other antibiotics such as rifampicin, to nutrients[46], and even pharmaceuticals[47].

## Methods

**Strains and media**. *E. coli* Top10 was used for all cloning. For characterization, we used *E. coli* strain AB360 (ref. [14]) for strains containing rTetR plasmids and/or T7RNAP plasmids. The strain contains the transcription factor AraC, whereas arabinose-metabolizing genes *araBAD* are deleted, and *lacYA177C* that allows for titratable arabinose regulation. SKA684 (*MG1655 ΔaraCBAD ΔlacIZYA ΔaraE ΔaraFGH attB::lacYA177C ΔrhaSRT ΔrhaBADM Tn7::tetR kan(FRT)*), a kanamycin-resistant SKA703 (ref. [48]) variant was used for experiments involving the TetR repressor. Plasmids were transformed using a one-step preparation protocol of competent *E. coli* for transformation of plasmids in testing strains[49].

Autoclaved LB-Miller medium was used for strain propagation. Sterile-filtered M9 medium supplemented with 0.2% casamino acids, 0.4% glucose, 0.001% thiamine, 0.00006% ferric citrate, 0.1 mM calcium chloride, and 1 mM magnesium sulfate was used for all gene expression experiments. Antibiotics (Sigma-Aldrich Chemie GmbH) were used as necessary for plasmid maintenance at concentrations of 100 μg/mL ampicillin, 34 μg/mL chloramphenicol, and 50 μg/mL kanamycin. Arabinose was received from Sigma-Aldrich Chemie GmbH. aTc was received from Chemie Brunschwig AG.

**Plasmids and genetic parts**. As a reporter plasmid for TetR- or rTetR-regulated expression, we used a pZ-series plasmid[30] containing a chloramphenicol resistance gene and a pSC101 origin of replication. The resulting plasmid pAB269 encodes mCherry under control of a $P_{tac}$ promoter[50] followed by a *tetO2* operator sequence and an RBS (see genetic parts list in Supplementary Data 1).

rTetR was derived from rtTA3 (pLenti CMV rtTA3, a gift from Eric Campeau, Addgene plasmid # 26429; http://n2t.net/addgene:26429; RRID:Addgene_26429). We used amino acids 1–207 from rtTA3, under control of $P_{araB*}$ (ref. [14]) and containing an ampicillin resistance gene and a colEI origin of replication to create pAB286.

**Growth and light incubation conditions**. All experiments were performed in an environmental shaker (INNOVA 40 R, New Brunswick) at 37 °C with shaking at 230 r.p.m. and black, clear bottom 24-well plates (Dunn Labortechnik GmbH, #303008), which was sealed with one layer of pierced polyolefin foil (HJ-BIOA-NALYTIK GmbH, Art. Nr. 900371) to reduce liquid evaporation, and an additional layer of gas-permeable membrane (BREATHseal™ Greiner Bio-One GmbH) foil to guarantee sterility, as well as a plastic lid (Greiner Bio-One GmbH, Product #: 656171). For experiments, overnight cultures in M9 medium and grown overnight to an $OD_{600}$ of ~4. aTc was added to 50 ng/ml for overnight cultures containing rTetR and grown in light-proof black tubes (Greiner Bio-One) to reduce accumulation of mCherry in overnight cultures. These cultures were diluted 1:20,000 into fresh M9 medium containing the respective inducer concentrations, right before the start of the experiment. This high dilution ensures that the cells are still in logarithmic growth phase after 5 h, at the end of the experiment[14] (Supplementary Fig. 10). One ml of inoculated culture was incubated per well of the 24-well plates.

For light induction experiments, we used a 24-well light plate apparatus (LPA)[34] equipped with UVA (375 nm, LEDsupply, PART #: L7-0-U5TH15-1), blue (472 nm, Super Bright LEDs Inc., Part #: RL5-B2545), green (525 nm, Super Bright LEDs Inc., Part Number: RL5-G8045), and infrared (740 nm, Marktech Optoelectronics, Part #: MTE1074N1-R) LEDs. The LEDs were calibrated (Photometer: Thorlabs PM100USB with S170C) to the LED with the lowest intensity of the respective wavelengths, resulting in the following calibrated maximal intensities: UVA 4.1 ± 0.5 mW/cm², blue 4.8 ± 0.3 mW/cm², green 1.9 ± 0.1 mW/cm², and infrared 0.70 ± 0.01 mW/cm². Light intensity correlated linearly with the input number given to the LPA via its program IRIS (Supplementary Fig. 11).

Cells were grown for 5 h before transcription, and translation was stopped with rifampicin and Tc[14]. A total of 100 μL inhibition solution was aliquoted into 96-well U-bottom plates (Thermo Scientific Nunc), precooled on ice, and samples were added in equal volumes (100 μL), resulting in a final inhibitor concentration of 250 μg/mL rifampicin (Sigma-Aldrich Chemie GmbH) and 25 μg/mL Tc (Sigma-Aldrich Chemie GmbH). The inhibition solution contained 500 μg/mL rifampicin and 50 μg/mL Tc in phosphate-buffered saline (Sigma-Aldrich Chemie GmbH, Dulbecco's phosphate-buffered saline), and was filtered using a 0.2 μm syringe filter (Sartorius). After sample was added, the solution was incubated on ice for at least 30 min. Then mCherry maturation was carried out at 37 °C for 90 min. The samples were kept at 4 °C until measurement through flow cytometry. Due to the slow degradation of aTc with blue, green, and infrared light, we preincubated the M9 medium with aTc without cells. After this incubation, we inoculated the preincubated media and grew the cells for 5 h before inhibition.

**Flow cytometry measurement**. Cell fluorescence was characterized using a Cytoflex S flow cytometer (Beckman Coulter) equipped with CytExpert 2.1.092 software. mCherry fluorescence was measured with a 561 nm laser and 610/20 nm bandpass filter, and following gain settings: forward scatter 100, side scatter 100, mCherry gain 1500 when mCherry was expressed from aTc-regulated promoters, and 300 gain when mCherry was expressed with Opto-T7RNAP due to the difference in expression levels. Thresholds of 2500 FSC-H and 1000 SSC-H were used for all samples. The flow cytometer was calibrated before each experiment with QC beads (CytoFLEX Daily QC Fluorospheres, Beckman Coulter) to ensure comparable fluorescence values across experiments from different days. At least 15,000 events were recorded in a two-dimensional forward and side scatter gate, which was drawn by eye and corresponded to the experimentally determined size of the testing strain at logarithmic growth and was kept constant for analysis of all experiments, and used for calculations of the mean and CV using the CytExpert software (Supplementary Fig. 12).

**Cell growth measurement using flow cytometry**. Cell growth was determined as the ratio of cells per defined volume of counting particles (2 μm AccuCount Blank Particles). For this, 79 μL of 50 μg/ml Tc and 500 μg/mL rifampicin (Sigma-Aldrich Chemie GmbH) in phosphate-buffered saline for transcription and translation inhibition, and 21 μL of AccuCount Blank Particles (Spherotech) for counting reference was used per sample. The cell solution was added in equal volumes (100 μL cell culture to 100 μL of inhibition and counting solution). Absolute cell counts per particle counts were determined by flow cytometry. For this, samples were measured for 2 min at 10 μL/min. The gating of the AccuCount particles and cells is shown in Supplementary Fig. 12 bottom. The reported data are from three biological replicates pooled from experiments performed on the same day. The doubling time of E. coli was measured to be 36.9 ± 1.2 min in our setup (Supplementary Fig. 13). We further determined that an inoculum of 1:20,000 of overnight culture into fresh media leads to an $OD_{600}$ of 0.034 ± 0.003 after 5 h incubation, therefore ensuring logarithmic growth throughout the experiments. Further, a calibration of optical density of a cell culture ($OD_{600}$) to our cell growth measurement using flow cytometry (counts/bead) was performed, which shows a linear correlation until an $OD_{600}$ of 0.2 (Supplementary Fig. 14).

**Mathematical modeling**. The TetR aTc dose–response (Fig. 1c) was fitted to a Hill-like equation of the following form:

$$f_{TetR}(x) = a + r_{max}\frac{x^n}{k_m + x^n},\qquad(1)$$

where $f_{TetR}(x)$ describes the gene expression controlled by TetR as a function of aTc concentration, $x$ represents aTc concentration, $a$ corresponds to the basal promoter activity in fully repressed conditions, $(a + r_{max})$ is the maximal promoter expression, $k_m$ is aTc's dissociation constant for TetR, and $n$ is the Hill coefficient for TetR. This dose–response was used subsequently to obtain aTc concentration estimates from the fluorescence readouts.

The measured aTc degradation responses to light intensity and light duration (Fig. 2b) were fitted to exponential decay Eq. (2)

$$f(u) = e^{-cu},\qquad(2)$$

where $u$ represents the input (light intensity or light duration), and $c$ is the decay factor. The data were normalized to have a maximal value of one prior to the fitting. Thus, the output of such a function lies in the range of [0,1], and represents the fraction of aTc degraded due to the applied input.

The fitted mathematical models for light duration and intensity-dependent aTc degradation, with TetR as sensor, were used to predict the response of rTetR to a set of inputs. As the values outputted by the fitted mathematical models correspond to the fraction of aTc degraded and not to aTc concentrations, these were multiplied by the aTc initial concentration to make the predictions.

The aTc dose–response curve of rTetR (Fig. 1f) was fitted to a Hill equation of the following form:

$$f_{rTetR}(x) = r_{max} - \frac{(r_{max} - a)x^n}{k_m + x^n},\qquad(3)$$

where $f_{rTetR}(x)$ describes the gene expression controlled by rTetR as a function of aTc concentration, $x$ represents aTc concentration, $r_{max}$ corresponds to the maximal promoter expression, $a$ is the promoter activity in fully repressed conditions, $k_m$ is aTc's dissociation constant for rTetR, and $n$ is the Hill coefficient for rTetR.

The aTc dose–response of the bandpass circuit (Fig. 4a) was empirically modeled combining the dose responses of aTc to TetR Eq. (1) and rTetR Eq. (3) as follows:

$$f_{prp}(x) = \frac{x}{x + k},\qquad(4)$$

$$f_{BP}(x) = f_{TetR}(x)(1 - f_{prp}(x)) + f_{rTetR}(x)f_{prp}(x),\qquad(5)$$

where $f_{prp}(x)$ describes the proportionality factor that determines how much $f_{TetR}(x)$ and $f_{rTetR}(x)$ is considered as a function of aTc concentration, $x$ represents aTc concentration, and $k$ defines the concentration of aTc, where $f_{TetR}$ and $f_{rTetR}$ are considered in equal proportions; $x$ represents aTc concentration. $f_{BP}(x)$ describes the gene expression controlled by the TetR–rTetR bandpass depending on the aTc concentration. We chose this empirical model over a mechanistic model as it required to fit only one parameter $k$ to the bandpass circuit experimental data (Fig. 4a), while parameters of $f_{TetR}(x)$ and $f_{rTetR}(x)$ were not modified.

All data were fitted using a nonlinear least squares optimizer (MATLAB R2015a 8.5.0.197613, MathWorks) and all parameter values are shown in Supplementary Table 1.

**Reporting summary**. Further information on research design is available in the Nature Research Reporting Summary linked to this article.

## Code availability
Code used for simulations is available on reasonable request from the corresponding author. Source data are provided with this paper.

## Data availability
The authors declare that the data supporting the findings of this study are available within the paper and its Supplementary Information files, and are available from the authors upon reasonable request. Sequences of novel plasmids as well as genetic parts used to modify published plasmids are shown in Supplementary Data 1. Source data are provided with this paper.

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

## Acknowledgements
The authors thank Umesh Shahdadpuri for assistance in preliminary characterization of blue light-dependent aTc characterization. The authors further thank Peter Buchmann for help with building the LPA. We also thank Dr. Stephanie Aoki for comments on the manuscript and helpful discussions. A.B. is part of the Life Science Zurich graduate school. This project has received funding from the European Research Council (ERC) under the European Union's Horizon 2020 research and innovation programme (CyberGenetics; grant agreement 743269), and FET-Open research and innovation actions grant under the European Union's Horizon 2020 research and innovation programme (CyGenTiG; grant agreement 801041).

## Author contributions
A.B. conceived the project, designed and performed all experiments, and analyzed the data. M.R. performed the mathematical modeling with assistance from A.B. M.K. supervised the project and provided funding. A.B. wrote the manuscript with contributions from M.R. and M.K. All authors have given approval to the final version of the manuscript.

## Competing interests
The authors declare no competing interests.

**Additional information**

