## [Peer Review File · Nature Communications]

Reviewers' Comments:

Reviewer #1:

Remarks to the Author:

Summary

This manuscript introduced a novel dynamic cellular control approach utilizing the photosensitivity of aTc and Tc, with easy control over active molecules level using UVA light, accompanied with a mathematical model of the light dependency. In addition to describing this new regulator system, the author also showcased its application in control of gene expression, growth and reducing dark state activity, and improved the effective range of aTc through the construction of the chemical bandpass filter. This method could potentially be a valuable tool in optogenetic induction experiments and expand the focus of photosensitive molecules. The manuscript in general is well written with comprehensive analysis, but a few issues still need to be clarified/resolved.

Comments:

1. The authors showed that effective inactivation of aTc was achieved using UVA light - what about the phototoxic effect of UVA light on the other components of the experiments beside aTc(e.g. damage to other cellular macromolecules)? Does this limit the usage of this control system in certain experiments?
2. For all the figure captions, I suggest the authors break down and clearly list the captions of each panel, instead of lumping all the description in one paragraph in the caption. For example, I do not find a detailed description of 1D.
3. While the mathematical models can predict the aTc degradation separately in TetR and rTetR systems, can such prediction still be achieved in the concentration bandpass filter system to enable controllability of aTC inactivation?
4. While the majority of the analysis were focused on aTc, the last section on growth control utilized Tc instead, which makes the reader wonder -does Tc and aTc respond to UVA light similarly? Further discussion/data could facilitate future users of this system better design their experiments
5. In the same section discussion growth control using Tc, the statement " 2) that increasing Tc concentrations result in increasing growth rates" does not seem right. Did you mean increasing UVA exposure time instead of Tc concentration?
6. The figures were discussed out of order - Fig 4 was mentioned before Fig 3C and 3D. Consider reorganize the figures/or result sections.

Reviewer #2:

Remarks to the Author:

In their manuscript entitle "aTc and Tc as light-sensitive controllers – exploiting natural chemical photosensitivity for dynamic optogenetic set point control", Baumschlager and colleagues describe an often-overlooked property of bioactive small molecules: their photosensitivity. By degrading anhydrotetracycline with pulses of UV-A light, they show that the activation of a Tet-responsive promoter can be toggled from on to off or set to an intermediate level.

I have a very high opinion of the authors' optogenetic work to date, as well as their creativity in applying an industrial detoxification approach as a cell biology tool. However, I am a little doubtful about the possible utility of photo-degradation of aTc as an "optogenetic" approach.

This skepticism comes from a few sources: (1) that light-based control only goes 1 way (you

cannot reverse the process and recover functional aTc), (2) photodegradation and UV-A light can both be potent sources of toxicity for cells, and (3) Tet-responsive promoters can be tuned even more easily by simply varying the concentration of aTc applied to a population of cells, without the need to calibrate light intensity to the geometry of a particular system, the cell density and absorbance of the media, etc.

The authors may argue against (3) that their system enables acute removal of aTc, which is difficult to do otherwise (at least without a spin-down & resuspension in fresh media), but as they point out, we already have TetR and rTetR, so you can always use the 'reverse' repressor system to flip the sign of aTc addition (so that acute removal is replaced with acute addition, something that is very easy to achieve).

In summary: while the work is technically sound, my skepticism about the system's overall utility leads me to believe that the work does not rise to the level of general interest of Nat Comms & may be better suited for a more specialized synthetic biology journal.

Beyond these high-level concerns, I have some specific concerns regarding the authors' characterization of photo-degradation of aTc:

(1) Dense bacterial cultures tend to absorb visible light – this is the basis for the popular "optical density" measurements of cell density in liquid culture. Although OD at 600 nm is usually used to characterize density, liquid cultures are strong absorbers at shorter wavelengths, raising the possibility that all of the authors' measurements of light-to-aTc degradation are extremely cell density dependent (otherwise aTc in the middle of the culture will be hit by a lower light dose and thus protected from degradation).

(2) What is the chemical structure of the degradation product? This is actually a rather important consideration, because completely photoconverting the aTc in solution will lead to high concentrations of its photo-degradation products. We must be sure that these are not themselves biologically active, and it is worth addressing whether free radicals induced during photo-degradation might impair other cellular functions or invoke a stress response.

(3) Can the authors perform any studies to test for deleterious effects of UV-A radiation on cellular function? In addition to the possibility of degradation products negatively affecting cellular function, light in the 300-400 nm range is almost always accompanied by phototoxicity or DNA damage. It could be the case that the doses used here are too low to induce a large-scale cell stress response, but this must be validated with data (a UV dose response). Ideally, the authors could obtain UV dose responses to cell growth rate and stress pathway activation, both in the presence and absence of aTc.

Reviewers' comments:

Reviewer #1 (Remarks to the Author):

Summary

This manuscript introduced a novel dynamic cellular control approach utilizing the photosensitivity of aTc and Tc, with easy control over active molecules level using UVA light, accompanied with a mathematical model of the light dependency. In addition to describing this new regulator system, the author also showcased its application in control of gene expression, growth and reducing dark state activity, and improved the effective range of aTc through the construction of the chemical bandpass filter. This method could potentially be a valuable tool in optogenetic induction experiments and expand the focus of photosensitive molecules. The manuscript in general is well written with comprehensive analysis, but a few issues still need to be clarified/resolved.

We thank reviewer #1 for her/his positive comments and thoughtful suggestions to improve the manuscript.

Comments:

1. The authors showed that effective inactivation of aTc was achieved using UVA light - what about the phototoxic effect of UVA light on the other components of the experiments beside aTc (e.g. damage to other cellular macromolecules)? Does this limit the usage of this control system in certain experiments?

We understand the concern that the use of UVA light in an experimental setup raises questions around phototoxicity or unwanted side effects. However, the use of UVA, or even shorter wavelengths such as UVB-UVC, is established and common in optogenetics as well as for the activation of optochemical compounds.

In fact, the vast majority of photoswitchable or photocaged chemicals is triggered by UV light. Probably the biggest class of photoactivatable chemicals employs photolabile protecting groups.¹⁻⁴ Two widely used photocaging groups are o-nitrobenzyl- and coumarin-based protecting groups. The absorption maximum of, for example, photocleavable nitrobenzyl groups lies in the UVB-UVC range. Further, cis-trans isomerization of azobenzenes, another commonly used constituent of optobiology, is toggled between 410-450 nm and 300-350 nm light.⁵ Photoresponsive chelators, which were heavily employed in neurobiology before the rise of channelrhodopsins, used UV light to change the conformation between binding and unbinding.⁵⁻⁷

In addition, a widely and successfully used optogenetic protein (UVR8) responds to UVA-UVB. It is used as an optogenetic regulator in mammalian cells, and originates from an ultraviolet-B (280-315 nm) photoreceptor from *Arabidopsis thaliana*.⁸ Also in *Escherichia coli*, UV was previously employed for light control.⁹ One of the photoactivatable systems used originates from the UirS/UirR system¹⁰, a TCS from the cyanobacterium *Synechocystis sp.* PCC 6803 which is activated with UV (382-405 nm) and inactivated with green light. Membrane bound UirS binds UirR and sequesters it to the membrane in the dark state, which is liberated and phosphorylated by UV light. This system was transferred to *E.coli* where it enables binding to the *csiR1* promoter and transcription initiation with a dynamic range of about 4-fold.⁹

Another important consideration is that UVR8 as well as UirS, as protein-based regulators require recurrent light inputs to stay active, and photocaged chemicals can only be activated once, and not inactivated. However, our method has the advantage of combining both the dynamic feature of protein based optogenetic regulators and the minimal UV-light exposure of optochemical methods as light only needs to be administered when the setpoint needs to be changed.

In spite of aforementioned previous studies where UV light was used successfully, we agree that light-induced damage has to be considered for experimental setups.

To examine the possible impact of UV light in our setup, we carried out additional experiments in which we use growth rate as an indicator of cell stress. It has been shown that stress response triggered by reactive oxygen species, denaturation of macromolecules such as proteins, nucleic acids and lipids as well as temperature variation or osmotic fluctuation, cause a change in the cell state from growth-centric to a survival-centric.^{11,12} This can lead to decrease, plateau or even negative cell growth (cell lysis).¹¹ Since we perform our experiments with cells in log growth phase throughout the experiment, our setup is extremely sensitive to growth rate changes.

We designed 2 experiments, in which we inoculate media with an extremely low OD_{600} of about 0.00015, and split the culture into triplicates, which are treated with different durations of UVA exposure. We illuminate the cells with the maximal intensity used in this study, which is also the maximum possible with our light setup (4.1 mW/cm^2). In the first experiment (Figure 1A), we illuminate the cells for a given duration (0 to 30 min) at the beginning of the 5h incubation period, and measure the cell count as described in the materials and methods section. This by far exceeds the time which is necessary for inactivation of aTc concentrations for maximal induction (Manuscript Figure 2). As such a strong and long light input as shown in Figure 1A would not be required for full inactivation of standard aTc concentrations, we used a more realistic light-application scenario, in which light pulses are applied repeatedly throughout the whole 5h incubation period. We applied a light pulse every 5 min with increasing duration so that the total light illumination time is equivalent to the previous experiment, but equally spread out over the 5h incubation period (Figure 1B). Any change in the growth behavior, even if growth is just affected transiently, should be easily detectable due to the log growth function.

These controls show that UVA-illumination at the maximal intensity used in our study poses very minor effects and does not impair cell growth, even though our setup is extremely sensitive to growth rate changes.

Figure 1: Effect of UVA exposure on cell growth. (A) A UVA light pulse of varying duration with the maximum intensity of our light setup is applied at the beginning of a 5h incubation period. (B) UVA is applied in pulses of maximal intensity throughout the experiment, which are shown as the sum of the applied duration on the x-axis. For both experiments, absolute cell counts are determined through counting beads as described in the methods section of the manuscript. Diagrams show mean values and standard deviation of three biological replicates of cell counts normalized to counting beads for growth experiments measured after 5 h incubation time in all cases.

We included this discussion on these experiments into the main text of the manuscript as well as Fig.S3 of the Supplementary Information.

2. For all the figure captions, I suggest the authors break down and clearly list the captions of each panel, instead of lumping all the description in one paragraph in the caption. For example, I do not find a detailed description of 1D.

We have expanded the captions and clearly described each panel separately in the revised manuscript according to the reviewer's suggestion. This improved clarity of the presented figures.

3. While the mathematical models can predict the aTc degradation separately in TetR and rTetR systems, can such prediction still be achieved in the concentration bandpass filter system to enable controllability of aTc inactivation?

This is a good point. We want to emphasize that aTc degradation was modelled independently from the sensory molecules TetR and rTetR. Therefore, our mathematical analysis of aTc degradation does not rely on the sensory molecule (rTetR or TetR or a combination) it affects. We only used TetR

and rTetR to regulate a measurable output (mCherry) for which we independently fitted dose-response curves for each TetR- and rTetR-containing cells. We took the TetR readout for parameterization of the model for prediction of aTc degradation. We then used the rTetR dose-response for prediction of mCherry expression levels after UVA incubation and initial aTc concentrations, but kept the aTc degradation model as well as the parameters the same. This showed that we accurately predicted the aTc degradation behavior, which is independent from using rTetR or TetR. For exactly this reason, one can apply the mathematical prediction of the aTc concentration on any dose-response curve (e.g. also a fit of the aTc bandpass filter). To facilitate this, we constructed an empirical mathematical model of the aTc bandpass filter and added a description of the model into the Material and Methods section and with Figure S9 added a descriptive plot to the Supplementary Information. With only one parameter to calibrate, the model fits the data very well.

Since we did not perform light degradation experiments on the bandpass filter circuit, but still wanted to strengthen our claim that light-induced degradation is highly predictable, we also used our aTc degradation model to predict the mCherry output we obtained for our dynamic gene expression experiment (Manuscript Fig. 3A). We included these predictions in the modified manuscript Figure 3A and added them as black dots to the measured mCherry output in our dynamic gene expression experiment.

We have also expanded the figure caption of manuscript Figure 2, as well as in the corresponding paragraph of the main text in the manuscript to clarify that the aTc degradation was modelled independent of the aTc dose response curves. We think that this was not clearly stated before, and we therefore thank the reviewer for bringing this up.

4. While the majority of the analysis were focused on aTc, the last section on growth control utilized Tc instead, which makes the reader wonder -does Tc and aTc respond to UVA light similarly? Further discussion/data could facilitate future users of this system better design their experiments

We strongly focused on aTc in this work, as 1) in our view aTc has the most potential to be applied for transcriptional control and 2) because of faster inactivation of aTc compared to Tc, which might in part be due to the lower concentrations that are needed for full biological activity, and therefore allows one to exploit the dynamic property this method adds. We mentioned Tc as a proof of principle that this concept of exploitation of light-sensitivity goes beyond the inducer aTc, and in the Manuscript also refer to other light sensitive chemicals such as nutrients¹³, pharmaceuticals¹⁴ or also other antibiotics such as rifampicin. We give an example with Tc which fulfils a different biological function, but do not want to claim a full characterization which would be required for a direct comparison. However, due to the high structural similarities we expect similar inactivation mechanisms and properties for both aTc and Tc inactivation. A characterization of Tc, such as performed for aTc, is out of the scope of this study and should be addressed in future work. Further, Tc inactivation using UV light is commonly used in industrial settings as reviewer #2 correctly mentioned, but to our knowledge has not been applied to biological research. To facilitate further use and characterization, we have expanded the discussion with references to previous studies on UVA-inactivation of Tc¹⁵⁻¹⁸ for future design of experiments.

5. In the same section discussion growth control using Tc, the statement " 2) that increasing Tc concentrations result in increasing growth rates" does not seem right. Did you mean increasing UVA exposure time instead of Tc concentration?

Thank you for pointing this out. Indeed, we wanted to say "decreasing Tc concentrations result in increasing growth rate", which again is caused by increasing UVA exposure time, as the reviewer mindfully suggests. We have corrected this sentence in the manuscript.

6. The figures were discussed out of order - Fig 4 was mentioned before Fig 3C and 3D. Consider reorganize the figures/or result sections.

We have restructured the results section so that Fig 3C and 3D are mentioned before Fig 4.

Again, we want to thank reviewer 1 for his/her critical comments which led us to initiate further experiments to support the study, resulting in a significantly improved manuscript.

Reviewer #2 (Remarks to the Author):

In their manuscript entitled "aTc and Tc as light-sensitive controllers – exploiting natural chemical photosensitivity for dynamic optogenetic set point control", Baumschlager and colleagues describe an often-overlooked property of bioactive small molecules: their photosensitivity. By degrading anhydrotetracycline with pulses of UV-A light, they show that the activation of a Tet-responsive promoter can be toggled from on to off or set to an intermediate level.

I have a very high opinion of the authors' optogenetic work to date, as well as their creativity in applying an industrial detoxification approach as a cell biology tool. However, I am a little doubtful about the possible utility of photo-degradation of aTc as an "optogenetic" approach.

This skepticism comes from a few sources: (1) that light-based control only goes 1 way (you cannot reverse the process and recover functional aTc), (2) photodegradation and UV-A light can both be potent sources of toxicity for cells, and (3) Tet-responsive promoters can be tuned even more easily by simply varying the concentration of aTc applied to a population of cells, without the need to calibrate light intensity to the geometry of a particular system, the cell density and absorbance of the media, etc.

The authors may argue against (3) that their system enables acute removal of aTc, which is difficult to do otherwise (at least without a spin-down & resuspension in fresh media), but as they point out, we already have TetR and rTetR, so you can always use the 'reverse' repressor system to flip the sign of aTc addition (so that acute removal is replaced with acute addition, something that is very easy to achieve).

In summary: while the work is technically sound, my skepticism about the system's overall utility leads me to believe that the work does not rise to the level of general interest of Nat Comms & may be better suited for a more specialized synthetic biology journal.

We thank reviewer #2 for the thoughtful and critical assessment of our work.

Regarding point 1 we completely agree that in our set-up light-control only goes one way, while in typical optogenetic approaches, light-inducible proteins can also be inactivated with a second wavelength, or will revert to the uninduced state due to thermal reaction. While typical protein based optogenetic approaches have the advantage that light is the only input required, this comes with the disadvantage that such approaches rely on either constant illumination, or pulsed illumination with a defined frequency to maintain a certain activity level of the optogenetic regulator. This is inevitable, as all protein-based photosensors will revert to their dark state in a timeframe which is dependent on the individual parameters of the used regulator. This point is exactly what sets our approach apart from other optogenetic methods, and we see an advantage in this, because we do not rely on photosensory proteins (which show this dark-state reversion) for our regulation. Instead, we rely on a chemical inducer, which indeed brings the drawback that it needs to be added, but has the unprecedented advantage that it maintains the input – effectively as a setpoint. If systems do not need to be adapted continuously, but only at certain timepoints, this can be a major advantage. For example, in our recently published antithetic integral controller¹⁹ we used aTc as an inducer to create a disturbance at a certain timepoint for the controller device by adding aTc, and then observed the adaptation behavior of the system. This UVA inactivation approach would enable one to safely and quickly remove this disturbance, again through a single short light pulse, without causing additional global perturbations which would be inevitable if the culture would have to be washed and resuspended in new media. This is just one example out of many potential imaginable applications of this method.

Regarding point 2 we want to refer reviewer #2 to our response to reviewer #1 comment 1 (above) for general discussion on the use of UVA light in optobiology, and to the additional experiments we performed under the same comment. Very briefly, in our response we discuss that UVA is indeed commonly used in optobiology and we performed experiments on the effect of UVA illumination with our light setup to investigate its effect on the growth of the bacterial cultures.

Point 3: We agree with the reviewer that the aTc-induction system has the additional benefit that an inverse acting repressor (rTetR) was engineered based on the original TetR system. As the reviewer correctly mentions, rTetR allows one to "switch the sign" for aTc addition. Indeed we exploited this fact to expand the system in this study by combining these two repressor proteins to

develop the chemical bandpass filter. This system not only allows activation with lower concentrations of aTc, but it also lets one repress an activated system by adding even more aTc.

However, we disagree with the reviewer that *multiple* activation and inactivation cycles are possible by simply using rTetR in combination with TetR. Indeed, after one round of activation followed by repression, the system would reach its limit, and the addition of even more aTc will only lead to stronger repression. This is due to the fact that both TetR and rTetR respond to the same inducer. This is where UVA inactivation of aTc comes to the rescue. Indeed by using UVA degradation of aTc, *repeated* cycles of activation and inactivation can be achieved easily and practically. This enables us to expand the usability of the aTc inducer, making it possible to realize dynamic adjustments of aTc-controlled gene expression levels, which are currently practically not feasible.

Repeated cycles of activation and inactivation may also be achieved by repeatedly washing the cells, but this is not a satisfying solution for several reasons: 1) in many setups it is not practically possible 2) it imposes a strong disturbance to the environmental conditions of the cells; and 3) it is experimentally demanding.

Beyond these high-level concerns, I have some specific concerns regarding the authors' characterization of photo-degradation of aTc:

(1) Dense bacterial cultures tend to absorb visible light – this is the basis for the popular “optical density” measurements of cell density in liquid culture. Although OD at 600 nm is usually used to characterize density, liquid cultures are strong absorbers at shorter wavelengths, raising the possibility that all of the authors' measurements of light-to-aTc degradation are extremely cell density dependent (otherwise aTc in the middle of the culture will be hit by a lower light dose and thus protected from degradation).

We thank the reviewer for bringing up a point that will be of interest to some experimenters. We have conducted further experiments that show that aTc can also be inactivated in higher density cultures. For this, we use our rTetR strain to visualize activation of gene expression via inactivation of aTc. We inoculated the culture at an OD₆₀₀ of 0.097 ± 0.002 , added aTc to 25 ng/ml and incubated the culture as previously described for 5h at which time they reached an OD₆₀₀ of 3.283 ± 0.006 . We divided the culture at the beginning of the experiments and for triplicates, applied a 15 min light pulse at the beginning, or applied the same light input with a delay of 1, 2, 3 and 4 hours after of incubation. The results in Figure 2 show, that inactivation is indeed possible even at higher cell densities. If the pulse was applied later than after 3h, the expression level is lower, which is expected as also for log growth we saw that if induction was applied for less than 2h, the maximum expression was not achieved (see Supplementary Figure S10 for comparison). This effect is more pronounced in high optical densities, which might be because cells already transitioned from log growth to stationary growth phase (OD₆₀₀ 1.393 ± 0.015 after 3h incubation time). The histograms in Figure 3 further show unimodal activation even at high cell densities.

With this, we provide a proof of principle that our method is also applicable for denser cell cultures. Of course, we expect that due to absorption, light duration and intensity need to be adjusted appropriately according to the experimental conditions. However, this depends not just on cell culture density, but also media, light application path etc. of the specific experimental conditions that an experimenter would use this method for. It is neither feasible nor helpful if we would provide such characterizations, as they strongly depend on the beforementioned conditions, and could be misleading to potential users. However, they have to be performed by the experimenter themselves which simply means that the experimenter runs a characterization on the conditions that shall be used in an experiment if precise control of intermediate aTc levels are required. Nonetheless, we expect that in many experimental cases, on-off control will be desired. Therefore, light does not necessarily need to be applied precisely, but rather at a sufficient dose to inactivate all inducer molecules and toggle between on and off states. We added additional experiments, shown in Figure 4, in which we repeatedly toggle between setpoints of minimal and maximal expression.

In general, absorption of visible light by cell cultures has not only to be considered with the method for aTc degradation described in this manuscript, but needs to be considered for any approach involving light for activation/inactivation. It would be interesting to investigate in future studies, if mixing of small molecules such as aTc might even be advantageous in regard to uniformity of

expression in a cell culture compared to protein based optogenetic approaches, in which every cell has to see the same light input to respond similarly.

Figure 2: UVA inactivation at high cell densities. A pulse of UVA is applied to rTetR-containing cells at the start of incubation (timepoint 0) or after 1, 2, 3 and 4h of incubation for a total of 5h incubation time. Controls show the same culture without UVA (no UVA control) and without aTc addition (no aTc control).

Figure 3: Histograms of triplicates of the UVA experiment timepoints shown in Figure 2. UVA applied at the start (line 1, left 3 histograms), 1h (line 1, right 3 histograms), 2h (line 2, left 3 histograms), 3h (line 2, right 3 histograms), 4h (line 3 histograms) of the incubation. The diagram shows mean mCherry expression values and standard deviation of three biological replicates measured after 5h incubation. The histograms show mCherry expression values of individual samples measured after 5h incubation.

We included this discussion into the main text of the manuscript as well as Fig.S5 of the Supplementary Information as we think that the reviewer posed an interesting question which might be of interest to researchers.

Figure 4: On-off control using an “aTc toggle switch” through repeated addition and UVA inactivation of aTc using TetR-controlled mCherry expression. The inputs were applied subsequently for each data point shown, as previously described for the dynamic protein expression experiment shown in Fig.3A of the main manuscript. Starting from a culture without aTc in setpoint 1, aTc was alternately added to a concentration of 7.5 ng/ml, which already lies in the saturating expression regime for aTc, and removed via a 3 min UVA light pulse. The experiment was performed in duplicates (individual samples shown as grey diamonds) and the corresponding mean fluorescent value is depicted as black bars.

We included some discussion on this in the main text and added the experimental data to the Supplementary Information Fig.S6.

(2) What is the chemical structure of the degradation product? This is actually a rather important consideration, because completely photoconverting the aTc in solution will lead to high concentrations of its photo-degradation products. We must be sure that these are not themselves biologically active, and it is worth addressing whether free radicals induced during photo-degradation might impair other cellular functions or invoke a stress response.

Photodegradation of regularly used concentrations of aTc for full induction creates degradation products which are not active in regard to its original function as shown by our experiments. Further, we did not notice growth defects in these experiments. However, we agree with the reviewer, that multiple rounds of addition and photodegradation can lead to an accumulation of degradation products, and so it is good to know at which concentrations these degradation products will show a functional decrease or other negative effects on cells. Since our lab does not have the capability to analyze the chemical structure of the degradation product, we focused on the biological implications the product might have.

To address this concern, we performed an additional set of experiments in which we degraded high amounts of aTc, which go far beyond the concentrations we previously used, and by far exceed aTc concentrations necessary for full induction of TetR-based systems, to analyze the limits of our approach. We inactivated 200, 500, 1000, 1500 and 2000 ng/ml aTc with a 50 min UVA light pulse and then added media containing cells with an equal volume with the degradation products to effective concentrations of 100, 250, 500, 750 and 1000 ng/ml aTc degradation products.

We observe that functional aTc inactivation is achieved until a concentration of 1000 ng/ml and noticed only a slight decrease in the maximal expression output as shown in Figure 5. With concentrations higher than 750 ng/ml aTc, a decrease in growth rate was observed.

Regarding if the degradation products invoke other cellular functions of stress response, we want to refer to our response to comment 3, in which we use growth as a readout for stress and inactivate a high concentration of aTc in the presence of cells.

To summarize, UVA inactivation of aTc concentrations (500ng/ml), which allow for about 100-cycles full activation and inactivation of TetR (5ng/ml), shows only minor residual activity compared to its original function as an inducer for TetR, and no significant growth defects when inactivated in the presence of cells compared to the control with just UVA (Figure 6). We therefore set the safe limit

of this method under our experimental conditions to inactivation of 500 ng/ml. This threshold might depend on the sensitivity of the bacterial strain or other cells used, and should be evaluated by the experimenter before the use of the method, in the case that experiments require the addition and inactivation of very high amounts of aTc. We want to emphasize that for regular concentrations of aTc (e.g. 5-10 ng/ml) this means about 100 full inactivation cycles from full-on to full-off. However, although these results make the method very versatile, we presented this method as a way to enable set point control in which addition of aTc or UVA-induced removal allow one to change the setpoint to a certain level, and then the expression level stays at this level until a new input is applied. If, however, continuous high frequency adjustments of gene expression is required, which could lead to accumulation of even higher concentrations of degradation products than the ones tested here, we would suggest the use of light-inducible proteins with a fast dark-state reversal rate.

Figure 5: Inactivation of high aTc concentrations with UVA light. Media containing different concentrations of aTc was treated with a UVA pulse, before cells containing rTetR, which allows expression in the absence of aTc, were added to the final concentrations shown. The diagram shows mean mCherry expression values and standard deviation of three biological replicates measured after 5h incubation.

We included part of this discussion into the main text of the manuscript as well as Fig.S3 panel E of the Supplementary Information.

(3) Can the authors perform any studies to test for deleterious effects of UV-A radiation on cellular function? In addition to the possibility of degradation products negatively affecting cellular function, light in the 300-400 nm range is almost always accompanied by phototoxicity or DNA damage. It could be the case that the doses used here are too low to induce a large-scale cell stress response, but this must be validated with data (a UV dose response). Ideally, the authors could obtain UV dose responses to cell growth rate and stress pathway activation, both in the presence and absence of aTc.

We performed a UVA dose response either using constant light or pulsed light, as mentioned before in the response to reviewer 1 comment 1 to assess if a stress-response is induced with the conditions used in this study and found no indication for a large-scale stress response. We included a discussion on these experiments in the main text of the manuscript as well as Fig.S3 of the Supplementary Information.

As suggested by reviewer 2, we performed another UV dose response, again at conditions which dramatically exceed the duration that have to be used for our method, and in addition in the presence of very high aTc concentrations (300, 400 and 500 ng/ml). We used the maximum intensity of our light setup and up to 20 min of light exposure, and then incubated the cells for a total of 5h with the cells in log growth phase throughout the incubation time. For concentrations typically used for induction of the system (5 ng/ml for TetR close to full induction), only seconds to minutes of light pulses are required as shown in Figure 2 of our manuscript.

As previously mentioned, growth can serve as an indicator of cell stress and it has been shown that stress response triggered by reactive oxygen species, denaturation of macromolecules such as proteins, nucleic acids and lipids as well as temperature variation or osmotic fluctuation cause a change in the cell state from growth-centric to survival-centric.^{11,12} This can lead to decrease, plateau or even negative cell growth (cell lysis).¹¹ Since we perform our experiments with cells in log growth phase throughout the experiment, our setup is extremely sensitive to growth rate changes.

The growth rate was quantified as absolute numbers through flow cytometry measurement of cells/counting bead as described in the methods section. The medium was inoculated, and then split into subsamples to which aTc was added to different concentrations (0, 300, 400, 500 ng/ml). The stock solution of aTc was highly concentrated (100 µg/ml) so that the additional volume through

inducer addition can be neglected. Our results in Figure 6A show that samples containing high concentrations of aTc and controls without aTc appeared indistinguishable.

We also used our rTetR-containing strain to – in addition – be able to quantify the percentage of mCherry expressing cells with and without UVA incubation, and to show that even inactivation of high amounts such as 500 ng/ml and 20 min of continuous maximum intensity UVA light leads to aTc inactivation (Figure 6B) and unimodal expression (Figure 6C). The percentage of particles with a fluorescence level well above background >1000 mCherry a.u. was >98% for the P3 cell gate (see materials and methods for gating strategy for cell growth measurements). This value is identical to cells that were grown without UVA incubation. Residual <2% of events showing no fluorescence are unavoidable with our flow cytometry measurement setup. Given that the UVA incubation is performed at the beginning of the 5h incubation period, we would expect that high levels of DNA damage causing mCherry inactivation would be amplified due to cell division and therefore would have been visible as an elevated fraction of cells showing no or lower mCherry fluorescence. No such fractions were observed.

We included part of this discussion into the main text of the manuscript as well as Fig.S3 panel C,D and Fig.S4 of the Supplementary Information.

We want to thank reviewer 2 again for her/his critical feedback and suggestions to investigate the limits of our method. It initiated additional experiments which now clearly show that these limitations have a high margin, allowing for many rounds of aTc addition and removal, further strengthening our manuscript.

Figure 6: High aTc concentration inactivation in the presence of cells. (A) Inactivation of 300, 400 and 500 ng/ml with increasing duration of UVA illumination at maximal intensity in comparison with cultures containing no aTc. (B) mCherry expression level of 500 ng/ml aTc containing cultures incubated with increasing duration of UVA illumination at maximal intensity. (C) Histograms of mCherry expression with cells incubated with 500 ng/ml aTc (panel B) in triplicates without UVA incubation (row 1, left 3 histograms) or incubated with UVA light pulses of 1 min (row 1, right 3 histograms), 5 min (row 2, left 3 histograms), 10 min (row 2, right 3 histograms), 15 min (row 3, left 3 histograms), and 20 min (row 3, right 3 histograms). Diagrams show mean values and standard deviation of three biological replicates of mCherry values for gene expression experiments and cell counts normalized to counting beads for growth experiments measured after 5 h incubation time in all cases. The histograms show mCherry expression values of individual samples measured after 5h incubation.

References

1. Goeldner, M. & Eds, R. G. *Dynamic Studies in Biology Phototriggers, Photoswitches and Caged Biomolecules*. (Wiley-VCH Verlag GmbH & Co. KGaA, 2005).

2. Mathews, A. S., Yang, H. & Montemagno, C. Photo-cleavable nucleotides for primer free enzyme mediated DNA synthesis. *Org. Biomol. Chem.* **14**, 8278–8288 (2016).
3. Seo, T. S. *et al.* Photocleavable fluorescent nucleotides for DNA sequencing on a chip constructed by site-specific coupling chemistry. *Proc. Natl. Acad. Sci. U. S. A.* **101**, 5488–5493 (2004).
4. Meng, Q. *et al.* Design and synthesis of a photocleavable fluorescent nucleotide 3'-O-allyl-dGTP-PC-Bodipy-FL-510 as a reversible terminator for DNA sequencing by synthesis. *J. Org. Chem.* **71**, 3248–3252 (2006).
5. Lester, H. A. & Nerbonne, J. M. Physiological and pharmacological manipulations with light flashes. *Annu. Rev. Biophys. Bioeng.* **11**, 151–175 (1982).
6. Tsien, R. Y. New Calcium Indicators and Buffers with High Selectivity Against Magnesium and Protons: Design, Synthesis, and Properties of Prototype Structures. *Biochemistry* **19**, 2396–2404 (1980).
7. Gurney, A. M., Tsien, R. Y. & Lester, H. A. Activation of a potassium current by rapid photochemically generated step increases of intracellular calcium in rat sympathetic neurons. *Proc. Natl. Acad. Sci. U. S. A.* **84**, 3496–3500 (1987).
8. Crefcoeur, R. P., Yin, R., Ulm, R. & Halazonetis, T. D. Ultraviolet-B-mediated induction of protein-protein interactions in mammalian cells. *Nat. Commun.* **4**, 1779 (2013).
9. Ramakrishnan, P. & Tabor, J. J. Repurposing *Synechocystis* PCC6803 UirS-UirR as a UV-violet/green photoreversible transcriptional regulatory tool in *E. coli*. *ACS Synth. Biol.* acssynbio.6b00068 (2016). doi:10.1021/acssynbio.6b00068
10. Song, J. Y. *et al.* Near-UV cyanobacteriochrome signaling system elicits negative phototaxis in the cyanobacterium *Synechocystis* sp. PCC 6803. *Proc. Natl. Acad. Sci. U. S. A.* **108**, 10780–10785 (2011).
11. Tonner, P. D., Darnell, C. L., Engelhardt, B. E. & Schmid, A. K. Detecting differential growth of microbial populations with Gaussian process regression. *Genome Res.* **27**, 320–333 (2017).
12. Imlay, J. A. Pathways of Oxidative Damage. *Annu. Rev. Microbiol.* **57**, 395–418 (2003).
13. Off, M. K. *et al.* Ultraviolet photodegradation of folic acid. *J. Photochem. Photobiol. B Biol.* **80**, 47–55 (2005).
14. Doll, T. E. & Frimmel, F. H. Fate of pharmaceuticals - Photodegradation by simulated solar UV-light. *Chemosphere* **52**, 1757–1769 (2003).
15. Schwarzenbach, R. P., Gschwend, P. M. & Imboden, D. M. *Environmental organic chemistry*. (Wiley-Interscience, 2016).
16. Boreen, A. L., Arnold, W. A. & McNeill, K. Photodegradation of pharmaceuticals in the aquatic environment: A review. *Aquat. Sci.* **65**, 320–341 (2003).
17. Niu, J., Ding, S., Zhang, L., Zhao, J. & Feng, C. Visible-light-mediated Sr-Bi2O3 photocatalysis of tetracycline: Kinetics, mechanisms and toxicity assessment. *Chemosphere* **93**, 1–8 (2013).
18. Xu, L., Li, H., Mitch, W. A., Tao, S. & Zhu, D. Enhanced Phototransformation of Tetracycline at Smectite Clay Surfaces under Simulated Sunlight via a Lewis-Base Catalyzed Alkalinization Mechanism. *Environ. Sci. Technol.* **53**, 710–718 (2019).
19. Aoki, S. K. *et al.* A universal biomolecular integral feedback controller for robust perfect adaptation. *Nature* **570**, 533–537 (2019).

Reviewers' Comments:

Reviewer #1:

Remarks to the Author:

The authors have addressed my concerns.

Reviewer #2:

Remarks to the Author:

I thank the authors for their thoughtful reply and nice additional studies.

The authors have also convinced me that this system has distinct utility from the rtTA/tTA pair, and the new data estimating the # of cycles that can be carried out is very helpful in drawing this distinction further. (I agree that there is no way anyone would do 100 cycles of washout & addition, but could envision experiments involving 10+ cycles of UV clearance & aTc re-stimulation.)

Their quantification of growth/viability in the presence of high UV doses and photo-degraded aTc is also quite helpful, but it raises one additional important issue.

The authors use an assay involving cell counting normalized to spiked-in beads after 5 h from an initial OD of 0.0001, and while I am sure they did this work carefully, it is difficult to relate their measurements to standard estimates of bacterial growth rates (e.g. measuring a doubling time or growth curve) without more information.

Crucially, the reader cannot easily tell what OD the cultures have reached after 5 hours. I'm sure the authors agree that if the cultures made it to stationary phase, slow-growing ones may catch up to faster ones and any differences in growth rate would appear erroneously small, even if they had once been substantial. Notably, no slow-growing controls are included so that the accuracy of the method can be assessed, even though the authors assert that their method is sensitive.

A back of the envelope calculation suggests that this might be problematic. For a doubling time of 20 min, an initial OD of $1e-4$ and assuming OD increases linearly with cell density, 14 cell generations would be sufficient to get to an OD of 1.6 in ~ 4.67 hours. But of course this depends on media density and there may be a lag phase before log-phase growth begins.

Could the authors please report the OD of at least the control cultures after 5 h in the Figure S3 experiments? Better yet, a growth curve showing OD over the 5 h time period? If the ODs are still reasonably low & consistent with log phase growth (i.e., $OD \leq 0.5$), then I will be satisfied that UV-A and photo-degradation toxicity is unlikely to be an issue. On the off-chance they turn out to be in the neighborhood of stationary phase, I ask for these measurements to be re-done more carefully, with growth rates over time and additional slow-growing controls.

Reviewer #2 (Remarks to the Author):

I thank the authors for their thoughtful reply and nice additional studies.

The authors have also convinced me that this system has distinct utility from the rtTA/tTA pair, and the new data estimating the # of cycles that can be carried out is very helpful in drawing this distinction further. (I agree that there is no way anyone would do 100 cycles of washout & addition, but could envision experiments involving 10+ cycles of UV clearance & aTc re-stimulation.)

Their quantification of growth/viability in the presence of high UV doses and photo-degraded aTc is also quite helpful, but it raises one additional important issue.

The authors use an assay involving cell counting normalized to spiked-in beads after 5 h from an initial OD of 0.0001, and while I am sure they did this work carefully, it is difficult to relate their measurements to standard estimates of bacterial growth rates (e.g. measuring a doubling time or growth curve) without more information.

Crucially, the reader cannot easily tell what OD the cultures have reached after 5 hours. I'm sure the authors agree that if the cultures made it to stationary phase, slow-growing ones may catch up to faster ones and any differences in growth rate would appear erroneously small, even if they had once been substantial. Notably, no slow-growing controls are included so that the accuracy of the method can be assessed, even though the authors assert that their method is sensitive.

A back of the envelope calculation suggests that this might be problematic. For a doubling time of 20 min, an initial OD of $1e-4$ and assuming OD increases linearly with cell density, 14 cell generations would be sufficient to get to an OD of 1.6 in ~ 4.67 hours. But of course this depends on media density and there may be a lag phase before log-phase growth begins.

Could the authors please report the OD of at least the control cultures after 5 h in the Figure S3 experiments? Better yet, a growth curve showing OD over the 5 h time period? If the ODs are still reasonably low & consistent with log phase growth (i.e., $OD \leq 0.5$), then I will be satisfied that UV-A and photo-degradation toxicity is unlikely to be an issue. On the off-chance they turn out to be in the neighborhood of stationary phase, I ask for these measurements to be re-done more carefully, with growth rates over time and additional slow-growing controls.

The reviewer brings up an important point for evaluation of toxicity. It is indeed important that cultures remain in logarithmic growth phase throughout such experiments in order to make reliable statements about whether cell stress occurred or not. We have been aware of this, and stated in the material and methods section that the 1:20,000 dilution of overnight cultures guarantees that the cells were in logarithmic growth phase. As a basis for this assertion, we cited our previous work in which we evaluated the growth rate under the same conditions and showed the cells remain in logarithmic growth phase. However, it is true that we did not show experimental evidence in this manuscript that this is indeed the case.

The reviewer is correct that a doubling time of about 20 min can be reached by an *E. coli* culture under ideal conditions (e.g. LB medium no antibiotics and perfect culturing conditions). For example, a doubling time of 22.5 minutes was described in (Liang, Ehrenberg, Dennis, & Bremer, 1999) or 27 min was reported in (Campos et al., 2014) However, as described in the material and methods section, we use a minimal medium (M9) in our experimental setup to increase reproducibility of our experiments compared to complex media such as LB. In M9 bacterial growth is significantly slower. For example, doubling times of 42 min (Campos et al., 2014) and 38 min (Reshes, Vanounou, Fishov, & Feingold, 2008) have been reported. Assuming a doubling time of 38 minutes and a start OD of 0.0001 it would

require 10 doublings, or 380 minutes to reach an OD of 0.1024. So theoretically, our setup should ensure that the cells are in logarithmic growth phase throughout the experiment and that at the end of the experiment, an OD₆₀₀ of 0.1 should not be exceeded, which is significantly below the standard reference value of 0.5. To show that this is also practically the case, we performed several sets of experiments with the strain that was also used in Figure S3 for toxicity tests (AB859).

First, we repeated our protocol and evaluated the optical density at the end of the 5h incubation. For this, we grew two different overnight cultures. One culture in standard culture tubes, and another culture in a light-proof black tubes (Greiner Bio- One) to simulate slight differences in the overnight cultures. From these two overnight cultures, we inoculated two independent main cultures each as previously described with a 1:20,000 dilution. From these overnight cultures, we further also inoculated two independent main cultures each with a 1:10,000 dilution to simulate a large pipetting error, which should practically not occur as it requires double the inoculum. We then aliquoted triplicates of each of these cultures into a 24-well plate, treated exactly the same as all the other samples reported in this manuscript, and incubated the plate for 5h in the same incubator. **The OD₆₀₀ values obtained for the four cultures with 1:20,000 inoculum were 0.034 ± 0.002 , 0.037 ± 0.002 , 0.03 ± 0.001 , 0.034 ± 0.001 , and for the four cultures with 1:10,000 inoculum 0.061 ± 0.008 , 0.056 ± 0.001 , 0.075 ± 0.003 , 0.078 ± 0.004 .** These results show that our experimental setup ensures logarithmic growth throughout the experiment until inhibition and measurement of the cells and theoretically even allows for large errors in the inoculum.

To determine the doubling time under exactly our culture conditions, we inoculated three cultures of which culture 1 and 2 originate from one overnight culture and culture 3 originates from another independent overnight culture. As start OD we chose an OD₆₀₀ of 0.005 as this value lies above the detection limit of the instrument we use for measuring the optical density (Nanodrop™ 2000c, Thermo Scientific). We aliquoted the three independent cultures into 24-well plates, treat them exactly as done for all the experiments in this manuscript, and incubated the plate for 4h. Every 30 min, we remove one sample for each culture to measure the OD₆₀₀ and to take a sample for cell counting. We determined the doubling time to 37.7 min, 37.5 and 35.5 for cultures 1, 2 and 3 respectively (Figure 1). This value lies very close to previously discussed reported values for growth of *E.coli* in M9, therefore our previous mentioned calculations hold for our setup.

Figure 1: Growth curves of *E.coli* strain AB859. Strain AB859 was used for UVA and aTc inactivation toxicity tests shown in Figure S3. Shown are mean and standard deviation of technical replicates for OD600. To obtain accurate measurements of the growth rate throughout this experiment, a starting OD above our detection limit was used (OD600 of 0.005).

Next, to relate the values of cells/bead obtained in the previously performed growth experiments to ODs, we first inoculated two independent overnight cultures to an OD600 of 0.1 and grew them until an OD600 of approximately 0.5 (culture 1: OD600 0.438, culture 2 OD600 0.515) so that the cells have a log growth phase morphology, stopped cell growth on ice and serially diluted the two cultures in ice cold M9. We then measured the OD600 of these samples and aliquoted 100 μ L of cell solution with 100 μ L of our previously used inhibition solution containing the counting beads, which was subsequently measured through flow cytometry, resembling exactly the measurement setup from our previous growth experiments. The obtained results (Figure 2) show that counts/bead and OD600 show a linear correlation up to an OD600 of approximately 0.2. For higher ODs, the correlation becomes less reliable, because higher ODs than 0.2 cause abort rates of >5 % of events. For example, more than 10% of events were aborted for our samples in Figure 2 with OD600 of 0.223 and 0.261 and more than 20% of events were aborted for our samples with OD600 of 0.438 and 0.514. However, these regions are irrelevant for our measurements as we never cross an OD600 of 0.1 in our experiments. Further, for each OD dilution the OD600 was measured as technical triplicates and of the two cultures, separate triplicates were aliquoted to evaluate if pipetting errors might be of relevance for evaluation of growth. It is apparent from the horizontal error bars in Figure 2 bottom two panels, that pipetting errors do not play a role.

We can now use the linear correlation of OD600 to cells/beads to evaluate the cells/beads obtained in our previous experiments. Since the value of 200 counts/bead was never exceeded for the toxicity experiments shown in Figure S3 (refer to the source data file for non-normalized counts/bead), this means that all samples were well below an OD600 of 0.1 and therefore in exponential growth phase at the end of the experiment.

Figure 2: Correlation of OD600 to counts/bead. The top left and right show two independent cultures serially diluted and then the OD600 measured in technical triplicates, as well as three samples added to the inhibition solution containing counting beads (Materials and Methods section of the manuscript) for each dilution. Bottom panels show the OD600 regions which show a linear correlation of optical density to counts/bead. Shown are mean and standard deviation of technical replicates for OD600 and separated pipetted replicates for cell counting.

To summarize, we agree that the growth phase at the end of the experiment is very relevant for evaluation of toxicity experiments. This was considered for all of our experiments, as we have kept our cultures in exponential growth phase at all times. We have shown this first based on literature of growth rates of *E.coli* with the media we used, and in addition with the control experiments described here. We have added parts of this discussion to the Manuscript (page 18) and the Supplementary Information (Figure S13 and Figure S14).

We thank the reviewer for the thoughtful and detailed evaluation of our manuscript, and we hope that with these experiments the reviewer agrees that UV-A and photo-degradation toxicity is not an issue for our method.

Reviewers' Comments:

Reviewer #2:

Remarks to the Author:

I am completely satisfied and very excited to see this paper published. Thanks to the authors for their careful, excellent work.